# Beyond Cost-to-go Estimates in Situated Temporal Planning

**Andrew Coles,[2] Shahaf S. Shperberg,[1] Erez Karpas, [4]**
**Solomon E. Shimony[1] Wheeler Ruml,[3]**
[1]Ben-Gurion University, Israel; [2]King's College London, UK;
[3]University of New Hampshire, USA; [4]Technion, Israel;
andrew.coles@kcl.ac.uk, shperbsh@post.bgu.ac.il, karpase@technion.ac.il
shimony@cs.bgu.ac.il, ruml@cs.unh.edu

## Abstract

Heuristic search research often deals with finding algorithms for offline planning which aim to minimize the number of expanded nodes or planning time. In online planning, algorithms for real-time search or deadline-aware search have been considered before. However, in this paper, we are interested in the problem of *situated temporal planning* in which an agent's plan can depend on exogenous events in the external world, and thus it becomes important to take the passage of time into account during the planning process. Previous work on situated temporal planning has proposed simple pruning strategies, as well as complex schemes for a simplified version of the associated metareasoning problem. In this paper, we propose a simple metareasoning technique, called the crude greedy scheme, that can be applied in a situated temporal planner. Our empirical evaluation shows that the crude greedy scheme outperforms standard heuristic search based on cost-to-go estimates.

## Introduction

For many years, research in heuristic search has focused on the objective of minimizing the number of nodes expanded during search (e.g (Dechter and Pearl 1985)). While this is the right objective under various scenarios, there are various scenarios where it is not. For example, if we still want an optimal plan but want to minimize search time, selective max (Domshlak, Karpas, and Markovitch 2012) or Rational Lazy A* (Karpas et al. 2018) can be used. Other work has dealt with finding a boundedly suboptimal plan as quickly as possible (Thayer et al. 2012), or with finding any solution as quickly as possible (Wilt and Ruml 2015). Departing from this paradigm even more, in motion planning the setting is that edge-cost evaluations are the most expensive operation, requiring different search algorithms (Mandalika, Salzman, and Srinivasa 2018; Haghtalab et al. 2018).

While the settings and objectives mentioned above are quite different from each other, they are all forms of offline planning. Addressing online planning raises a new set of objectives and scenarios. For example, in real-time search, an agent must interleave planning and execution, requiring still different search algorithms (Koenig and Sun 2009; Sharon, Felner, and Sturtevant 2014; Cserna, Ruml, and Frank 2017; Cserna et al. 2018). Deadline-aware search (Dionne, Thayer, and Ruml 2011) must find a plan within some deadline. The

BUGSY planner (Burns, Ruml, and Do 2013) attempts to optimize the utility of a plan, which depends on both plan quality and search time.

In this paper we are concerned with a recent setting, called *situated temporal planning* (Cashmore et al. 2018). Situated temporal planning addresses a problem where planning happens online, in the presence of external temporal constraints such as deadlines. In situated temporal planning, a plan must be found quickly enough that it is possible to execute that plan after planning completes. Situated temporal planning is inspired by the planning problem a robot faces when it has to replan (Cashmore et al. 2019), but the problem statement is independent of this motivation.

The first planner to address situated temporal planning (Cashmore et al. 2018) used temporal reasoning (Dechter, Meiri, and Pearl 1991) prune search nodes for which it is provably too late to start execution. It also used estimates of remaining search time (Dionne, Thayer, and Ruml 2011) together with information from the temporal relaxed planning graph (Coles et al. 2010) to estimate whether a given search node is likely to be *timely*, meaning that it is likely to lead to a solution which will be executable when planning finishes. It also used dual open lists: one only for timely nodes, and another one for all nodes (including nodes for which it is likely too late to start execution). However, the planner still used standard heuristic search algorithms (GBFS or Weighted A*) with these open lists, while noting that this is the wrong thing to do, and leaving for future work finding the right search control rules.

Inspired by this problem, a recent paper (Shperberg et al. 2019) proposed a rational metareasoning (Russell and Wefald 1991) approach for a simplified version of the problem faced by the situated planner. The problem was simplified in several ways: first, the paper addressed a one-shot version of the metareasoning problem, and second, the paper assumed distributions on the remaining search time and on the deadline for each node are known. The paper then formulated the metareasoning problem as an MDP, with the objective of maximizing the *probability* of finding a timely plan, and showed that it is intractable. It also gave a greedy decision rule, which worked well in an empirical evaluation with various types of distributions.

In this paper, we explore using such a metareasoning approach *as an integrated part of* a situated temporal planner.

This involves addressing the two simplifications described above. The naive way of addressing the first simplification — the one-shot nature of the greedy rule — is to apply it at every expansion decision the underlying heuristic search algorithm makes, in order to choose which node from the open list to expand. The problem with this approach is that the number of nodes on the open list grows very quickly (typically exponentially), and so even a linear time metareasoning algorithm would incur too much overhead. Thus, we introduce an even simpler decision rule, which we call the *crude greedy* scheme, which does not require access to the distributions, but only to their estimated means. Additionally, the crude greedy scheme allows us to compute one number for each node, $\hat{Q}$, and expand nodes with a high $\hat{Q}$-value first. This allows us to use a regular open list, although one that is not sorted according to cost-to-go estimates, as in standard heuristic search. In fact, as we will see, cost-to-go estimates play no role in the ordering criterion at all.

An empirical evaluation on a set of problems from the Robocup Logistics League (RCLL) domain (Niemueller, Lakemeyer, and Ferrein 2015; Niemueller et al. 2016) shows that using the crude greedy scheme in the situated temporal planner (Cashmore et al. 2018) leads to a *timely* solution of significantly more problems than using standard heuristic search, even with pruning late nodes and dual open lists. Next, we briefly survey the main results of the metareasoning paper (Shperberg et al. 2019), and then describe how we derive the crude greedy decision rule, and conclude with an empirical evaluation that demonstrates its efficacy.

# The metareasoning MDP and practical approximations

A model called AE2 ('allocating effort when actions expire') that assigns processing time under the simplifying assumption of $n$ independent processes was proposed by Shperberg et al. (2019). In order to make this paper self-contained, we re-state the model and its properties below.

## The AE2 Model

The AE2 model abstracts away from the search, and assumes $n$ processes (e.g., each process can be thought of as a node on the open list) that each attempts to solve the same problem under time constraints. (For example, these may represent promising partial plans for a certain goal, implemented as nodes on the frontier of a search tree, but as discussed below the problems may be completely unrelated to planning.) There is a single computing thread or processor to run all the processes, so it must be shared. When process $i$ terminates, it will, with probability $P_i$, deliver a solution or, otherwise, indicate its failure to find one. For each process, there is a deadline, defined in absolute wall clock time, by which the computation must be completed in order for any solution it finds to be valid, although that deadline may only be known to us with uncertainty. For process $i$, let $D_i(t)$ be the CDF over wall clock times of the random variable denoting the deadline. Note that the actual deadline for a process is only discovered with certainty when its computation is complete. This models the fact that, in planning, a dependence on an external timed event might not become clear until the final action in the plan is added. If a process terminates with a solution before its deadline, we say that it is *timely*. The processes have performance profiles described by CDFs $M_i(t)$ giving the probability that process $i$ will terminate given an accumulated computation time on that process of $t$ or less. Although some of the algorithms we present may work with dependent random variables, we assume in our analysis that all the variables are independent. Given the $D_i(t)$, $M_i(t)$, and $P_i$, the objective of AE2 is to schedule processing time over the $n$ processes such that the probability that at least one process finds a solution before its deadline is maximized. This is the essential metareasoning problem in planning when actions expire.

## The Deliberation Scheduling MDP

We now represent the AE2 problem of deliberation scheduling with uncertain deadlines as a Markov decision process. For simplicity, we initially assume that time is discrete and the smallest unit of time is 1. Allowing continuous time is more complex because one needs to define what is done if some time-slice is allocated to a process $i$, and that process terminates before the end of the time-slice. Discretization avoids this complication.

We can now define our deliberation scheduling problem as an the following MDP, with distributions represented by their discrete probability function (pmf). Denote $m_i(t) = M_i(t) - M_i(t-1)$, the probability that process $i$ completes after exactly $t$ time units of computation time, and $d_i(t) = D_i(t) - D_i(t-1)$, the probability that the deadline for process $i$ is exactly at time $t$. Without loss of generality, we can assume that $P_i = 1$: otherwise modify the deadline distribution for process $i$ to have $d_i(-1) = 1 - P_i$, simulating failure of the process to find a solution at all with probability $1 - P_i$, and multiply all other $d_i(t)$ by $P_i$. This simplified problem we call SEA2. We formalize the SEA2 MDP as an indefinite duration MDP with terminal states, where we keep track of time as part of the state. (An alternate definition would be as a finite-horizon MDP, given a finite value $d$ for the last possible deadline.)

The actions in the MDP are: assign the next time unit to process $i$, denoted by $a_i$ with $i \in [1, n]$. We allow action $a_i$ only if process $i$ has not already failed.

The state variables are the wall clock time $T$ and one state variable $T_i$ for each process, with domain $\mathcal{N} \cup \{F\}$. $T_i$ denotes the cumulative time assigned to each process $i$ until the current state, or that the process has completed computation and resulted in failure to find a solution within the deadline. We also have special terminal states SUCCESS and FAIL. Thus the state space is:

$$\mathcal{S} = (dom(T) \times \bigtimes_{1 \leqslant i \leqslant n} dom(T_i)) \cup \{\text{SUCCESS, FAIL}\}$$

The initial state is $T = 0$ and $T_i = 0$ for all $1 \leqslant i \leqslant n$.

The transition distribution is determined by which process $i$ has last been scheduled (the action $a_i$), and the $M_i$ and $D_i$ distributions. If all processes fail, transition into FAIL with probability 1. If some process is successful, transition into SUCCESS with probability 1. More precisely:

- The current time $T$ is always incremented by 1.

- Accumulated computation time is preserved, i.e. for action $a_i$, $T_j(t+1) = T_j(t)$ for all processes $j \neq i$.

- $T_i(t) = F$ always leads to $T_i(t+1) = F$.

- For action $a_i$ (assign time to process $i$), the probability that process $i$'s computation is complete given that it has not previously completed is $P(C_i) = \frac{m_i(T_i+1)}{1-M_i(T_i)}$. If completion occurs, the respective deadline will be met with probability $1 - D_i(T_i)$. Therefore, transition probabilities are: with probability $1 - P(C_i)$ set $T_i(t+1) = T_i(t) + 1$, with probability $P(C_i)D_i(T_i)$ set $T_i(t+1) = F$ (process $i$ failed to meet its deadline), and otherwise (probability $P(C_i)(1 - D_i(T_i))$) transition into SUCCESS (the value of $T_i$ in this case is 'don't care').

- If $T_i(t+1) = F$ for all $i$, transition into FAIL.

The reward function is 0 for all states, except SUCCESS, which has a reward of 1.

## Solving the AE2 Model

It was shown in Shperberg et al. (2019) that solving the AE2 MDP is NP-hard, and it was conjectured to be even harder (possibly even PSPACE-complete, like similar MDPs). On the other hand, under the restriction of known deadlines and a special condition of diminishing returns (non-decreasing logarithm of probability of failure) that an optimal schedule can be found in polynomial time. However, neither known deadlines nor diminishing returns strictly hold in practice in planning processes. Still, the algorithm for diminishing returns provided insights that were used to create an appropriate greedy scheme. The greedy scheme, briefly repeated below, is relatively easy to compute and achieved good results empirically.

Define $m_i(t) = M_i(t) - M_i(t-1)$, the probability that process $i$ completes after exactly $t$ time units of computation time. Under an allocation $A_i = (0,t)$ in which all processing time starting from time 0 until time $t$ is allocated to process $i$, the success distribution for process $i$ is:

$$f_i(t) = PS_i(A_i = (0,t)) = P_i \sum_{t'=0}^{t} m_i(t')(1-D_i(t')) \quad (1)$$

Define the *most effective computation time* for process $i$ under this assumption to be:

$$e_i = \operatorname*{argmin}_t \frac{log(1 - f_i(t))}{t} \quad (2)$$

The latter is justified by observing that the term $-log(1 - f_i(t))$ behaves like utility, as it is monotonically increasing with the probability of finding a timely plan in process $i$; and on the other hand it behaves additively with the terms for other processes. That is, if we could start all processes at time 0 and run them for time $t$, and if all the random variables were jointly independent, then indeed maximizing the sum of the $-log(1 - f_i(t))$ terms results in maximum probability of a timely plan.

However, since not all processes can start at time 0, the intuition from the diminishing returns optimization is thus

to prefer the process $i$ that has the best utility per time unit. i.e. such that $-log(1 - f_i(t))/(e_i)$ is greatest. Still, allocating time now to process $i$ delays other processes, so it is also important to allocate the time now to a process that has a deadline as early as possible, as this is most critical. Shperberg et al. (2019) therefore suggested the following greedy algorithm: Whenever assigning computation time, allocate $t_d$ units of computation time to process $i$ that maximizes:

$$Q(i) = \frac{\alpha}{E[D_i]} - \frac{log(1 - f_i(e_i))}{e_i} \quad (3)$$

where $\alpha$ and $t_d$ are positive empirically determined parameters, and $E[D_i]$ is the expectation of the random variable that has the CDF $D_i$, which we use as a proxy for 'deadline of process $i$'. (This is a slight abuse of notation in the interest of conciseness, as $E[D_i]$ could be taken to mean the expectation of the CDF, which is *not* what we want here.) The $\alpha$ parameter trades off between preferring earlier expected deadlines (large $\alpha$) and better performance slopes (small $\alpha$).

## Improved Greedy Scheme

Using the proxy $E[D_i]$ in the value $Q(i)$ is reasonable, but somewhat ad-hoc. It also encounters problems if $E[D_i]$ is zero or even near-zero. A more disciplined scheme can indeed use the utility per time unit as in $Q(i)$, but the first term should be better justified theoretically. The reason for including the deadline in $Q(i)$ is in order to give preference to processes with an early deadline, because deferring their processing may cause them to be unable to complete before their deadline (even if they *would* have been timely had they been scheduled for processing immediately). Therefore, instead of the first term it makes sense to provide a measure of the "utility damage" to a process $i$ due to delaying its processing start time from time 0 to time $t_d$. This can be computed exactly, as follows. Define a 'generalized' $f_i'$, the probability of process $i$ finding a timely plan given a contiguous computation time $t$ starting at time $t_d$, as follows:

$$f_i'(t, t_d) = PS_i(A_i = (t_d, t)) = P_i \sum_{t'=0}^{t} m_i(t')(1-D_i(t'+t_d)) \quad (4)$$

Note that this is the same as $f$, except that processing starts at time $t_d$, which is the same as saying that the deadline distribution is advanced by $t_d$ (and indeed, $f_i(t) = f_i'(t, 0)$).

Assuming that the time we wish to assign to process $i$ is $e_i$, before the delay we can achieve a utility of: $-log(1 - f_i(e_i))$, and after delay of $t_d$ can achieve $-log(1 - f_i'(e_i, t_d))$. The difference between the former and the latter values is the 'damage' caused by the delay. Thus, our improved greedy scheme is to assign $t_d$ time units to the process that maximizes:

$$Q'(i) = \alpha[log(1-f_i'(e_i, t_d))-log(1-f_i(e_i))] - \frac{log(1 - f_i(e_i))}{e_i} \quad (5)$$

Observe that the first term is proportional to the logarithm of:

$$\frac{1 - f_i'(e_i, t_d)}{1 - f_i(e_i)} \quad (6)$$

## Integrating the greedy scheme into a planner

In order to actually use the greedy scheme in a planner, several issues must be handled. Foremost is the issue of obtaining the distributions, which is non-trivial. Second, although the greedy scheme is quite efficient, it is not quite efficient enough for making decisions about node expansions, which must be done in essentially negligible time. Hence, we consider a crude version of the greedy scheme below.

## Crude version of the greedy scheme

| Id | $h$ | Crude Greedy with $\alpha =$ | | | | | |
|----|------|--------|------|------|------|------|--------|
| | | $-10^4$ | $-1$ | $0$ | $0.1$ | $1$ | $10^4$ |
| 1 | 3.61 | 0.45 | 0.47 | 0.45 | 0.85 | 0.45 | 0.59 |
| 2 | 13.45 | 1.91 | 1.81 | 1.78 | 2.52 | 1.77 | x |
| 3 | x | 1.75 | 1.74 | 1.61 | 2.24 | 1.65 | 5.14 |
| 4 | x | 1.03 | 1.04 | 1 | 1.41 | 1.02 | 1.19 |
| 5 | 9.42 | 0.59 | 0.51 | 0.57 | 0.73 | 0.49 | 0.4 |
| 6 | - | - | - | - | - | - | - |
| 7 | x | 10.89 | 9.74 | 9.63 | 17.55 | 9.97 | x |
| 8 | x | x | 3.57 | 3.77 | 5.98 | 3.88 | 2.04 |
| 9 | x | 2.36 | 2.46 | 2.5 | 3.16 | 2.4 | 2.27 |
| 10 | 0.66 | 0.37 | 0.37 | 0.38 | 0.49 | 0.38 | 0.36 |
| 11 | 0.28 | 0.24 | 0.24 | 0.25 | 0.31 | 0.27 | 9.05 |
| 12 | 0.31 | 0.25 | 0.3 | 0.24 | 0.34 | 0.24 | 0.17 |
| 13 | 0.9 | 0.44 | 0.45 | 0.46 | 0.55 | 0.45 | 0.43 |
| 14 | 11.88 | 1.55 | 1.64 | 1.6 | 2.11 | 1.59 | 1.19 |
| 15 | 1.54 | x | x | x | x | x | 0.59 |
| 16 | x | 2.33 | 2.27 | 2.29 | 3.56 | 2.29 | 1.71 |
| 17 | x | 1.59 | 1.55 | 1.54 | 2.52 | 1.59 | 3.93 |
| 18 | x | 2.27 | 2.27 | 2.35 | 4.34 | 2.29 | x |
| 19 | x | 1.61 | 1.6 | 1.62 | 2.73 | 1.6 | 6.6 |
| 20 | - | - | - | - | - | - | - |
| 21 | x | x | x | x | x | x | 1.71 |
| 22 | 0.76 | 0.43 | 0.43 | 0.41 | 0.42 | 0.48 | 1.37 |
| 23 | x | 2.06 | 2.33 | 1.99 | 2.04 | 2.36 | 1.78 |
| 24 | 14.67 | 1.55 | 1.61 | 1.49 | 1.63 | 1.92 | 0.52 |
| 25 | 0.57 | 0.31 | 0.32 | 0.28 | 0.28 | 0.35 | 0.46 |
| 26 | 4 | x | x | x | x | x | x |
| 27 | x | 1.92 | 1.88 | 1.75 | 1.72 | 1.82 | x |
| 28 | x | 0.52 | 0.74 | 0.49 | 0.47 | 0.47 | 0.48 |
| 29 | 50.03 | x | 1.21 | 1.13 | 1.13 | 1.23 | 0.56 |
| 30 | - | - | - | - | - | - | - |
| 31 | 2.48 | x | x | x | x | x | 0.96 |
| 32 | x | 1.74 | 1.79 | 1.56 | 1.91 | 1.74 | 1.59 |
| 33 | x | x | x | x | x | x | x |
| 34 | 3.37 | 0.54 | 0.63 | 0.54 | 0.27 | 0.59 | 0.26 |
| 35 | 2.73 | x | x | x | x | 0.34 | 0.23 |
| 36 | 10.18 | 0.75 | 0.76 | 0.62 | 0.57 | 0.87 | 0.71 |
| 37 | - | - | - | - | - | - | - |
| 38 | - | - | - | - | - | - | - |
| 39 | 1.09 | 0.54 | 0.65 | 0.48 | 0.49 | 0.52 | 0.49 |
| 40 | - | - | - | - | - | - | - |
| 41 | 1.28 | 0.27 | 0.28 | 0.27 | 0.52 | 0.28 | 0.24 |
| 42 | 1.32 | x | x | x | x | x | 0.42 |
| SOLVED | 21 | 27 | 29 | 29 | 29 | **30** | **30** |

Table 1: Planning Time on RCLL Instances

Consider Equation 3 defining $Q(i)$. The estimate for $E[D_i]$ in the first term can use any current estimate of the deadline time. For the second term in $Q(i)$, we can approximate $e_i$ by the expected time to return a solution. We use estimate both of these quantities as described by Cashmore et al. (2018). We now briefly review these estimates, but refer the interested reader to the original paper.

To estimate the current deadline $E[D_i]$, we use the temporal relaxed planning graph (TRPG) (Coles et al. 2010). Specifically, we compute the *slack* of the chosen relaxed plan, that is, how much we can delay execution of the entire plan (the actions leading to the current node together with the actions in the relaxed plan). Note that, because the relaxed plan is not guaranteed to be optimal, this is not necessarily an admissible estimate.

To estimate the remaining search time $e_i$, we use an idea from Deadline Aware Search (Dionne, Thayer, and Ruml 2011). We estimate the 'distance from solution' (i.e. estimation of number of expansions from the current node, also based on the relaxed), and divide it by the 'progress rate' (i.e. the reciprocal of the time difference between the time a node is expanded and the time its parent was expanded, averaged over multiple nodes).

The numerator $\log(1 - f_i(e_i))$ is more problematic, as it requires $f_i$, which uses the complete distribution. Note that this term is negative, and we want it to be as large as possible in absolute value. The simplest crude approximation is a constant $\log(1 - f_i(e_i))$, but that is an oversimplification. Note that if the most effective computation time $e_i$ is greater than $E[D_i]$ then in fact we are not likely to find a solution in time in process $i$. For simplicity, we thus use $-\log(1 - f_i(e_i)) \approx max(0, \beta(E[D_i] - e_i))$ for some parameter $\beta$ as a first approximation. The idea here is that $E[D_i] - e_i$ is an estimate of the *slack* (spare time) we have in completing the computation before the deadline. We are then assuming that the negative logarithm of the probability of not completing in time is approximately proportional to the slack. This slack is also already estimated by the situated temporal planner, based on the partial plan to the current node and the temporal relaxed planning graph from it (Cashmore et al. 2018).

Note that once we plug this into Equation 3, the $\beta$ can be absorbed into the $\alpha$ parameter. An additional issue is that in a planner, since $E[D_i]$ is relative to the time now, this value keeps decreasing and may approach 0. This may cause the $\frac{\alpha}{E[D_i]}$ term to grow without bound. To fix this, we bound the denominator away from 0 to the time $t_{10}$, the time required for 10 node expansions.

In summary, for our crude greedy approach, we expand next the node with the highest value of

$$\hat{Q}(i) = \frac{\max(0, E[D_i] - e_i)}{e_i} + \frac{\alpha}{\max(E[D_i], t_{10})} \quad (7)$$

This crude version of the greedy scheme has two advantages: it does not require the complete distributions $D_i$ and $M_i$, and is more computationally efficient as it does not have to compute the summation in the equation for $f_i$. This comes at the cost of a potential oversimplification that may cause schedule quality to excessively degrade. An additional problem is that the original greedy scheme itself using the $E[D_i]$ was only a first-order approximation, and in fact distributions can be devised where it fails badly.

## Empirical Results

To evaluate the crude greedy scheme, we implemented it on top of the situated temporal planner of Cashmore et al. (2018), which itself is implemented on top of OPTIC (Benton, Coles, and Coles 2012). We ran the planner using the crude greedy scheme, with different values of $\alpha$, and compared it to the original situated temporal planner, which sorts its open lists based on cost-to-go estimates (denoted $h$ below). Both planners used exactly the same pruning method for nodes which are guaranteed to be too late, and the same dual open list mechanism for preferring nodes which are likely to be timely.

We compared the results of the different planners on instances of the Robocup Logistic League Challenge (Niemueller, Lakemeyer, and Ferrein 2015; Niemueller et al. 2016), a domain that involves robots moving workpieces between different workstations. The goal is to manufacture and deliver an order within some time window, and thus situated temporal planning is very natural here. Table 1 shows the planning time for the baseline planner ($h$) and the planner using the crude greedy scheme with different values of $\alpha$. In the table, 'x' means 'failed to find a plan in time to satisfy the deadline(s)'. As these results show, the crude greedy scheme solves significantly more problems than the baseline for any value of $\alpha$. This provides support for a metareasoning approach to allocating search effort in situated planning. It also suggests that, for situated temporal planning, cost-to-go estimates are not the right primary source of heuristic guidance.

## Conclusion

In this paper, we have provided the first practical metareasoning approach for situated temporal planning. We showed empirically that this approach outperforms standard heuristic search based on cost-to-go estimates. Nevertheless, the temporal relaxed planning graph (Coles et al. 2010) serves an important purpose here, allowing us to estimate both remaining planning time and the deadline for a node. Thus, we believe our results suggest that cost-to-go estimates are not as important for situated temporal planning as they are for minimizing the number of expanded nodes or planning time as in classical heuristic search.

The metareasoning scheme we provided is a crude version of the greedy scheme of Shperberg et al. (2019). We introduced approximations in order to make the metareasoning sufficiently fast and in order to utilize only readily available information generated during the search. We also proposed a more refined and better theoretically justified version of the algorithm ('improved greedy'), but making the improved version applicable in the planner is a non-trivial challenge that forms part of our future research.

### Ongoing Work: Crude version of the improved greedy scheme

The improved greedy scheme is better justified, but has an additional term where we need the complete distribution ($f'(t, t_d)$ is needed, rather than just the expectation $E[D_i]$).

We would like to replace this distribution with a small number of parameters than can be easier to obtain. Basically the same considerations apply here as well, except that the the term involving $f_i'$ requires access to the full distributions $m_i$, $D_i$. Given specific distribution types, it may be possible to compute this term as a function of $E[D_i]$ and $e_i$. However, this part of the work is still in progress and at present we are not sure what parameters we can obtain during the search that would support the improved scheme.

## Acknowledgements

Partially supported by ISF grant #844/17, and by the Frankel center for CS at BGU. Project also funded by the European Union's Horizon 2020 Research and Innovation programme under Grant Agreement No. 821988 (ADE).

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
