# OpenReview forum: "Beyond Cost-to-go Estimates in Situated Temporal Planning"
_icaps-conference.org/ICAPS/2019/Workshop/HSDIP_

### Official Review · AnonReviewer2 · 2019-04-04
**Interesting problem, encouraging results**

**Rating:** 7
**Confidence:** 3

**Review:**

The paper addresses the problem of situated temporal planning, in which the planning agent needs not only to compute a plan, but to do so while taking into account the time it takes to compute that plan, as plans might have e.g. deadlines, real clock times where the validity of the plan expires.

The paper builds on previous work by Shperberg et al. (2019), where a simplification of the above problem is cast as an MDP, its NP-hardness is shown, and some greedy strategies proposed. The current paper proposes a further simplification on these greedy strategies, dubbed the crude greedy scheme, with the aim of making it feasible (i.e. fast enough) to implement this strategy in an actual situated planner. The paper then evaluates the strategy on some instances of the Robocup Logistic League Challenge, showing that it outperforms the approach in (Shperberg et al. 2019).

The line of research addressed in the paper is interesting and very relevant for the HSDIP community, and I understand the contribution as a first attempt at making the previous ideas practical to implement in a planner. The results on the Robocup Logistics benchmark are certainly encouraging, and overall I found the paper well written. I would therefore recommend for acceptance.

Below I outline a few comments / concerns that perhaps the authors could address in the final version of the paper or on the presentation:

- I would appreciate some discussion on the relation between the AE2 model and the actual search procedure being modeled. As far as I understand, when modeling a search procedure, P_i models the probability that a solution is found in the subtree rooted at node i. Is this correct? If this is the case, assumption that P_i’s are independent seems a rather strong assumption.

- Related to the above, it is not fully clear to me how you bridge the gap between the idealized AE2 model and the actual implementation of the search procedure. Perhaps it would be good to spell out in full detail how the different components of  $\hat{Q}$ end up being estimated (for instance, I read (p.3, right col) the details of how e_i is estimated, but I am still not sure how the "estimation of number of expansions from the current node" is done; and similar for the "slack")

- As far as the empirical evaluation is concerned, it is hard to understand what is the impact of the \alpha parameter on the results. Perhaps the authors could share some intuition or data that helps understanding it?

- typo (page 1): used temporal reasoning (Dechter...) *to* prune search nodes.

---

> ### Author Response · Authors · 2019-04-10
> **Clarifications and answers**
>
> Pi were assumed independent as a simplifying assumption, as we believed that
> we could not hope to obtain data for a more refined model with dependencies,
> and also it makes the MDP even harder than currently (NP-hard without dependencies!)
> In fact in the actually implemented crude greedy we are not even using Pi at all,
> we  could not obtain sufficient online information to use it.
>
> Bridging the model with the implementation: will be elaborated if accepted.
>
> Alpha: will be further elaborated, roughly as follows: alpha zero means we go by
> "contribution slope" only, ignoring deadlines. alpha=10000 means we consider almost
> only the deadline. Large negative alpha was a "sanity test", meaning prefer LOWER contribution slope and is expected  to do badly (which it did).

---

### Official Review · AnonReviewer1 · 2019-04-05
**Interesting experimental observation, but significance and contribution not clear**

**Rating:** 5
**Confidence:** 4

**Review:**

Very interesting topic, but not clear exactly what is the contribution or the message the authors want to convey.

 Key details of the AE2 model are not defined in the paper. The authors say that it is an MDP but basic facts about it are not present in the manuscript. This is a major omission, and made quite difficult to follow the description of the approximations being proposed.

The background the authors give needs to be better organised and the notation reviewed for consistency. For instance, D_i is defined as a time variant function, but then the argument is discarded. Also the authors define D_i as a CDF, and then use the expectation of the CDF (E[D_i]), which is the component of the formulation crucial to justify the approximations proposed later on. It is mathematical fact that the CDF of a random variable has a uniform distribution (as per the Probability Integral Transform), but D_i is not bounded, which means that E[D_i] is not well defined (it can be infinite). Bounding D_i in turn implies that the approximation described in the last subsection before the experimental results is quite arbitrary, since the expected value of a uniform distribution over the interval [a,b] is 1/2 (a+b).

The experimental results seem to suggest that in the set of instances considered by the authors, what seems to me to be essentially random search, is outperforming temporal planners based on forward heuristic search. That is interesting, and the experimental observation that follows is valuable.

But, how significant it is? Without needing to consider other benchmarks, the experimental design in the paper would have been improved significantly if rather than just one trial, several trials had been reported, with different realisations of the deadlines and giving confidence intervals on the runtimes. Would that make the observation go away? Or it would stay? That is a pretty basic question the authors should at least try to give an answer to.

Finally, I find surprising that a comparison of the makespans of the temporal plans computed by the heuristic search planner and the authors' is not presented.

Questions:

- What is the formal definition of a process? Is it a temporal action? That was my assumption, please correct me if I was wrong.

 - What are the state of the MDP like? How many decisions per stage need to be considered on average? Is it a stochastic shortest path (SSP) problem? How is transition function defined? Is it necessary to give conditional probability distributions over states and actions?

- What is the rationale for choosing t_10, the time to do 10 node expansions, for Equation 7? Why not 9? Or 11?

- In the last paragraph of the Section "Crude version of the greedy scheme" it is mentioned that f_i is computed from a convolution, but the operands (function) of the convolution are not described. Is "convolution" the mathematical operation the authors' are referring to?

- How does plan makespan compare with the ones produced by heuristic search?

---

> ### Author Response · Authors · 2019-04-10
> **Clarification of E[Di], MDP description, and other answers.**
>
> E[Di]: we apologize for this abuse of notation, but we did say that  this is the expected deadline, meaning it is
> the expected value of the random variable describing the location of the deadline, rather than
> "expectation of the CDF".
>
> The formal definition of the process, as well as the MDP description are defined in a previous work (Allocating Planning Effort when Actions Expire, Shperberg et al. 2019) on which this paper builds.
> We erroneously thought that the workshop had a 4pages+refs limit, and that we had no space for the MDP
> definition. If accepted, naturally we will add this. Meanwhile,  the gist of the MDP definition is:
>
> States variables: how much processing time was already allocated (and used) by each process i, for
> each process i whether it has completed and/or failed to find a solution(or succeeded to find a timely
> solution, which is a terminal "success" state, occurs if ANY process i delivers a timely plan).
> The state space is the (exponential-size) cross product of all the above state variables.
>
> Actions: Ai= assign 1 time unit now to process i.
>
> Reward on state: 0 except for "success" state which has reward 1
>
> Transitions: count up the timer for the process i for which time was allocated by Ai
> and process i  terminates with probability defined by Mi, and if it terminates,
> succeeds with probability determined by Pi, Di, and the real time elapsed.
> (Exact distributions in the cited paper).
>
> Rationale for choosing t10: this is the time used in the implementation between updates and
> recomputations of the Qi. We admitted in the paper this this is not well justified theoretically,
> hence the work (underway, as it is a workshop paper) for the improved greedy which does not
> have such ad-hoc constants, but which we do not yet know how to integrate inside a planner.
>
>
> Convolution: yes, this is a mathematical convolution.
>
> Makespan: not reported as it is not part of what we are optimizing, and completely do not care about
> in this paper (other than the part of the makespan which occurs before a deadline, in which case
> that part plus the planning time MUST be less then the time the deadline occurs (assuming NOW=0).

---

> > ### Comment · AnonReviewer1 · 2019-04-10
> > **Thanks for the response, need some time to process it**
> >
> > Thank you for acknowledging some of my comments. I will need some time to process the paper you're building on.
> >
> > If it was a convolution, which are the functions involved? And what is the rationale for it? Usually convolutions are used to smooth/filter signals. If you could elaborate on this I would appreciate it.

---

> > > ### Author Response · Authors · 2019-04-10
> > > **Specifics of convolution**
> > >
> > > We have:
> > >
> > > fi(t) = P_i \sigma_{t'=0}^t m_i(t')(1-D_i(t'))
> > >
> > > That is, we are convolving (a transformed) M_i with D_i, omitting the 0 terms,
> > > i.e. factoring in the fact that process i is not computing anything at t'>t and t'<0, so has 0
> > > probability of finding a solution at such times)
> > > Note that m_i (t) above is the probability mass function, i.e. m_i(t) = M_i(t) - M_i(t-1).
> > >
> > > Thus f_i(t) is the probability that process i will find a TIMELY solution (a solution before
> > > the deadline) if we start it NOW and allow it to run contiguously for t time units.

---

> > > > ### Comment · AnonReviewer1 · 2019-04-10
> > > > **That was helpful**
> > > >
> > > > Thank you very much, those details are appreciated and go a long way to provide clarity.

---

### Meta-Review · Program_Chairs · 2019-04-25

**Recommendation:** Accept
**Confidence:** 5

**Metareview:**

Dear Authors,
thank you very much for your submission. We are happy to inform you that
we have decided to accept it and we look forward to your talk in the workshop.
Please, go over the feedback in the reviews and correct or update your papers
in time for the camera ready date (May 24).
Best regards
HSDIP organizers